# Penguins Strike Back: A Report on the Unusual Case of Adélie Penguin (*Pygoscelis adeliae*) Attacks on South Polar Skua Nests Distant from the Breeding Colony

**Youmin Kim** [1,2,†]**, Jong-U Kim** [1,†]**, Hosung Chung** [1]**, Yeon-Soo Oh** [1]**, Young-Geun Oh** [1] **and Jeong-Hoon Kim** [1,*]

1   Division of Life Sciences, Korea Polar Research Institute, Incheon 21990, Korea; kym2603@kopri.re.kr (Y.K.); wildlife@kopri.re.kr (J.-U.K.); hchung@kopri.re.kr (H.C.); dhdustn960@kopri.re.kr (Y.-S.O.); oyg8195@kopri.re.kr (Y.-G.O.)
2   Department of Agriculture, Forestry, and Bioresources, Seoul National University, Seoul 08826, Korea
*   Correspondence: jhkim94@kopri.re.kr
†   These authors contributed equally.

**Keywords:** inter-specific behavior; *Pygoscelis adeliae*; Ross Sea region; *Stercorarius maccormicki*

Colonial seabirds use various methods to defend their nests from predators [1], such as mobbing behavior, emitting alarm calls, vomiting, and defecating [2]. In Antarctica, penguins aggressively defend their territory from skuas in order to guard their offspring, since skuas frequently prey on penguins' eggs or chicks [3–5]. Therefore, it has been reported that the degree of agonistic behavior was high around the nests and decreased when they were far from the colony [4,5]. On the other hand, Adélie penguins (*Pygoscelis adeliae*) sometimes crowd into skua nests located near the colony on the way to the sea and trample their eggs accidentally [6,7]. This occurrence could have a direct effect on the breeding success of sympatric skuas [6]. The egg loss of skuas caused by penguins was observed only around the penguin colony [8]; however, no such event has been reported yet in areas without a penguin colony.

Here, we present unusual photographs in which south polar skua (*Stercorarius maccormicki*) nests were attacked by Adélie penguins despite being far from the penguin colony. The pictures were taken by motion-capture cameras (solar trail camera, model solar and comparable; Spypoint, Springfield, USA) for monitoring the breeding performance of skuas at Cape Möbius (S 74°37′33″ E 164°13′08″), Ross Sea, during the 2019/2020 breeding season. The site is located within 1 km of the coast and 17 km away from Adélie Cove, where the nearest Adélie penguins' breeding colony (about 11,000 breeding pairs) is located (Figure 1) [9]. Additionally, the sea ice extended over 5 km from the adjacent coast during December 2019. Thus, we consider the presence of breeding Adélie penguins in these inland areas to be unexpected.

We found images of aggressive behavior by penguins toward skuas while checking thousands of photographs taken by the motion-capture cameras from three of eight monitored skua nests. Adélie penguins approached the nests and expelled the breeding skuas. In addition, one egg was broken in one of the nests (Figure 2A,B). According to the observations from the monitored nests, the Adélie penguins wandered around the eggs or a chick after evicting the parent skuas (Figure 2). These observations were made on 19, 26 December and 1 January, respectively, during the same period in which breeding penguins feed their chicks [10]. Although penguins threatened the adult skuas, they seemed to have no interest in the eggs or chicks of the skuas. Hence, the egg trampling might be an accidental event, which is supported by the successful fledging of all the skua offspring except for the egg broken by the penguins.

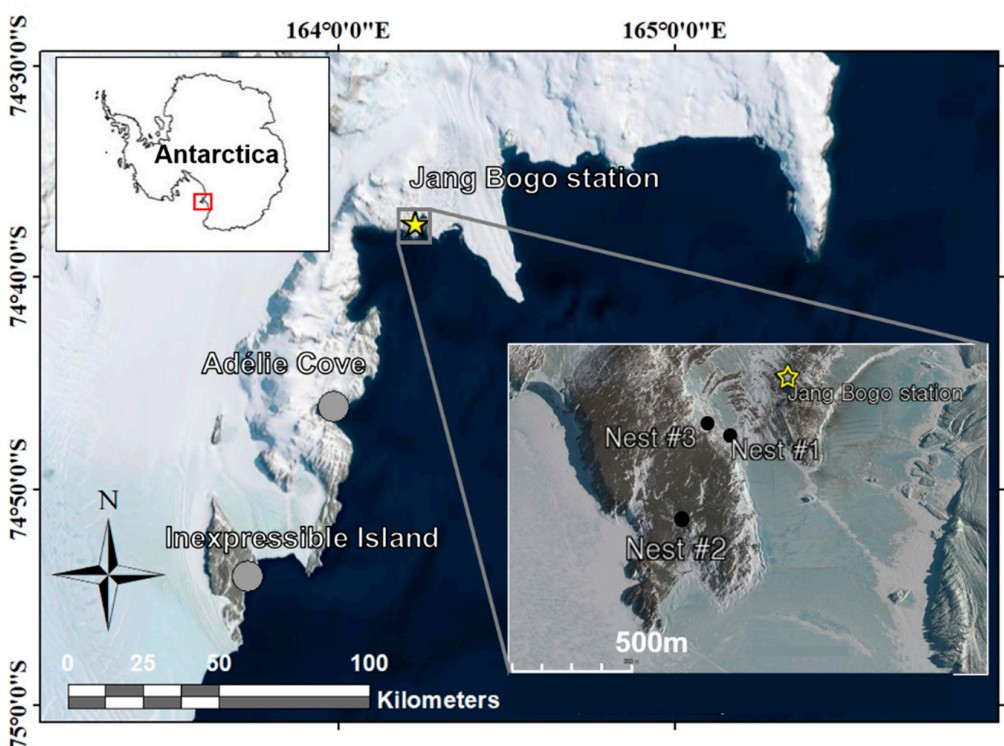

**Figure 1.** The location of the study site. The skua nest attack happened near the Jang Bogo station (Yellow star), Cape Möbius. The grey circle indicates the Adélie penguin colony, and the black circle indicates skua nests (The skua nests were marked with number and hash tag, underlying satellite image © Esri and Maxar Technologies).

To the best of our knowledge, this is a previously unreported interesting case in which the Adélie penguin attacked and caused egg loss at skua nests far from the penguins' colony. Although examples of Adélie penguins threatening or attacking the skuas' nests can be found on the internet and in research papers, all of these incidents were observed only around the penguin colony [6,7]. The aggressive behaviors of the Adélie penguins to breeding skuas are expected to have occurred elsewhere without penguin colonies, but this has not been reported yet. In this study, such cases were captured on motion-capture cameras for the first time. In general, Adélie penguins are more sensitive and aggressive when they are breeding [4], but the individuals attacking skua nests might be non-breeding or may have failed to breed [7]. Visiting skua nests located far from the colony during the chick-guarding periods may lead to disadvantages in rearing their own chicks and defending from predators or unexpected weather fluctuation; therefore, this suggests that the behavior observed at the study site may have been caused by a non-breeding flock that were roaming around the coastal area. The reason for the attack is unknown, but previous studies have speculated that the alarm calls to threaten their potential enemies from the incubating or brooding skuas stimulate the penguins' curiosity, making them approach the nest [6,7]. Although penguins broke a skua egg, when we checked the situation before and after the incident in a series of photos, they seemed to take interest in only adult skuas rather than the eggs. Only one skua egg was lost in the three nests, and we confirmed that all of the other eggs and chicks fledged with success. Therefore, the egg loss by Adélie penguins seems to be unintended, and our observation is an interesting case of showing aggression to the predator regardless of the prey's reproduction. Unusual inter-specific behavior and reciprocal interactions between two Antarctic top predator avian species observed from our survey will be helpful to enrich understanding of interactions among the biodiversity in the Ross Sea region. In addition, we expected that the motion-capture camera can improve observation of Antarctic seabirds without time limit.

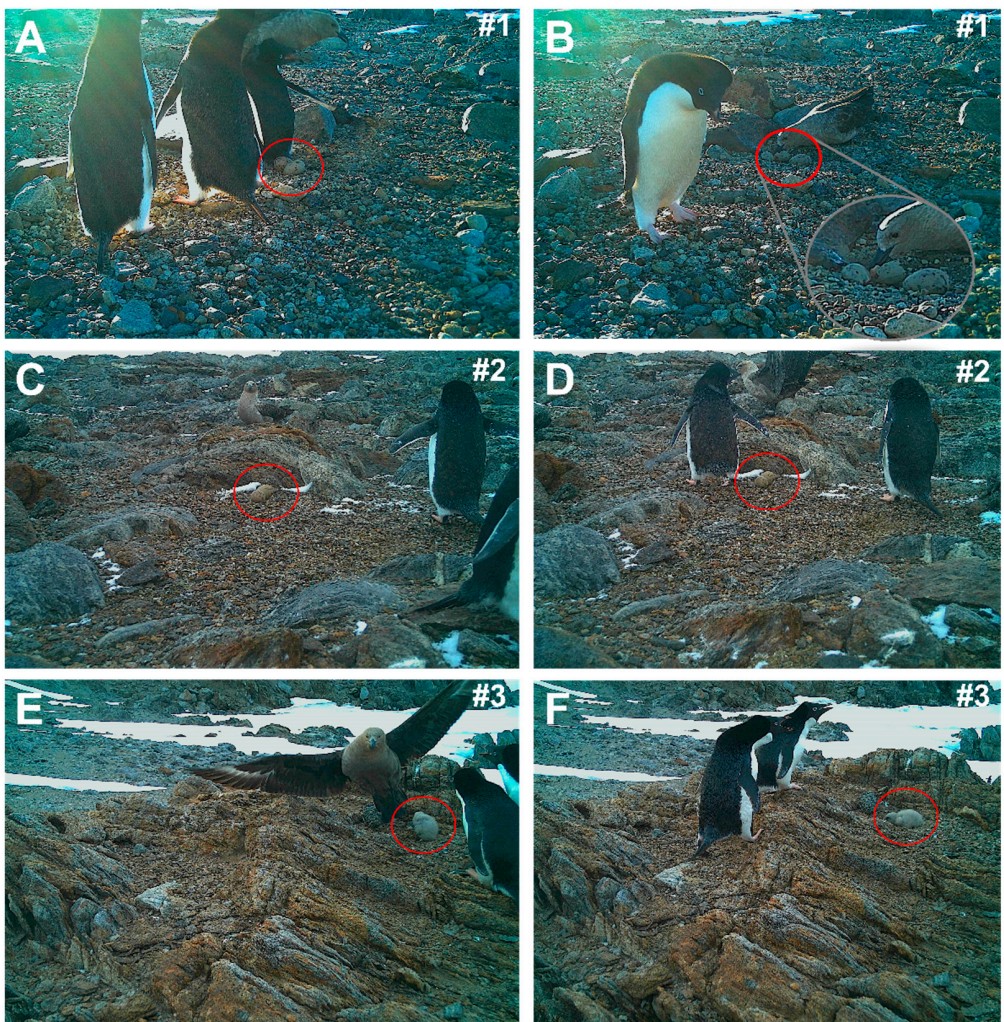

**Figure 2.** The photographs of Adélie penguins attacking three skua nests. (**A,B**) Adélie penguins approached a skua nest (Supplementary #1) and expelled an adult skua. One of the two eggs was observed to be broken in the next picture (26 December). (**C,D**) In another nest (Supplementary #2), penguins were passing near the nest of the skua, and they returned to chase the skua (19 December). (**E,F**) In the other nest (Supplementary #3), Adélie penguins expelled and threatened the parent skua (1 January). The red circle indicates the location of skua eggs or a chick. The photographs of three nests can be seen as gif files in the supplementary material (identified each nest by number and hash tag).

**Supplementary Materials:** The following are available online at https://www.mdpi.com/article/10.3390/d13050181/s1, Supplementary gif 1: Penguin attacked the skua nest #1. Supplementary gif 2: Penguin attacked skua nest #2. Supplementary gif 3: The behavior of penguins crossing near the skua nest #2. Supplementary gif 4: Penguin attacked skua nest #3.

**Author Contributions:** Conceptualization, J.-H.K. and J.-U.K.; methodology, J.-H.K. and Y.K.; validation, Y.K., J.-U.K., H.C., Y.-S.O., Y.-G.O. and J.-H.K.; investigation, J.-H.K. and J.-U.K.; data curation, Y.K., Y.-S.O. and Y.-G.O.; writing—original draft preparation, Y.K. and J.-U.K.; writing—review and editing, Y.K., J.-U.K., H.C., Y.-S.O., Y.-G.O. and J.-H.K.; supervision, J.-H.K.; project administration, J.-H.K. and H.C.; funding acquisition, J.-H.K. All authors have read and agreed to the published version of the manuscript.

**Funding:** This research was supported by the "Ecosystem Structure and Function of Marine Protected Area (MPA) in Antarctica" project (PM20060), funded by the Ministry of Oceans and Fisheries

(20170336), Korea, and "Study on polar ecosystem change by warming and adaptation mechanisms of polar organism" funded by the Korea Polar Research Institute (PE21140).

**Institutional Review Board Statement:** This study was conducted according to the permission from the Korean Ministry of Foreign Affairs in accordance with the Act on Antarctic Activities and Protection of Antarctic Environment. And this study carried out in accordance with the 'SCAR Code of Conduct for the use of Animals for Scientific Purposes in Antarctica.

**Informed Consent Statement:** Not applicable.

**Data Availability Statement:** The data is not publicly available due to its usage in the ongoing study.

**Acknowledgments:** We would like to thank Woo-Sung Kim for assistance with the fieldwork. The authors thank the logistic help from the overwintering members at Jang Bogo Station during the field seasons, especially Jong-Il Jeung who helped with the fieldwork.

**Conflicts of Interest:** The authors declare no conflict of interest.

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
