# Peer review of "Penguins Strike Back: A Report on the Unusual Case of Adélie Penguin (Pygoscelis adeliae) Attacks on South Polar Skua Nests Distant from the Breeding Colony"

_diversity, doi:10.3390/d13050181_

Round 1

Reviewer 1 Report

Review

In general, I believe this short report of agonistic behavior of Adélie penguins toward nesting South Polar skuas in areas far removed from penguin colonies is interesting, novel, and worthy of publication. The images provide clear evidence for an unanticipated interaction of this species pair. To improve this report for publication, I provide a few minor recommendations below.  One in particular is to abandon the concept of a reversal of predator-prey interactions, since the observation of aggressive behavior of penguins toward skuas is not equivalent to predation.

Specific comments

Line 16 – I suggest changing “show hostility to” to “aggressively defend their territory from”. Such a change would conform with the language of the citations regarding aggressiveness and defensive behaviors at the nest. Furthermore, such defensive behaviors at the nest contrast with the “offensive” behavior exhibited by the birds when visiting remote skua territories.

Line 31- consider adding the current population size at the Adélie Cove colony to help the reader understand why penguins would be unexpected visitors to Cape Möbius.

Line 31-32 – The point here, I gather, is to establish that you wouldn’t’ expect to find large groups of Adélie penguins so far from the coast. Consider saying as much. For example, “Thus, we consider the presence of Adélie penguins in these inland areas to be unexpected.”

Line 33 – delete “fascinating” and include somewhere in this sentence that you found images of aggressive behavior by penguins toward skuas

Line 36-37 – what do you mean “attracted to the nest”? On one hand, this could be correct that it was the nesting site that attracted the penguins, not the skua necessarily. But it seems difficult to support this assertion with only imagery. Rather, the imagery demonstrate the penguins remained in/near the nest after evicting the parent skuas. I suggest you change the text to reflect the direct observation.

Line 48 – suggest changing “violent and threatening” to “aggressive”

Line 52-53 – For the sake of clarity, I suggest deleting “but it was assumed that….[8]”. The reason is that you describe potential consequences of such remote visitation to inland skua nests by breeders in the next sentence and conclude that non-breeders are more likely to participate in the behavior you observed. Thus, the suggested deletion is redundant with your later text.

Line 58 - This hypothesis that Adélie penguins use the alarm calls of skuas to locate nests is interesting. Can you briefly clarify when and why skuas make alarm calls. For example, were the alarm calls only made when Adélie penguins were close, or are there other avian predators that the nesting skuas were vigilant toward? In other words, why are the skuas making alarm calls if it attracts Adélie penguins?

Line 64-65 – This concept of “reversal” suggested here is hard to accept, since the prey (penguins) did not become “predators” in a strict sense. The loss of eggs due to trampling cannot be construed as predation. The act of aggression by a typical prey species toward its predator, a kind of harassment, is not necessarily extraordinary, either. For example, consider mobbing of raptors by smaller species. I suggest revising this section in the context a reversed social interaction, such as mobbing or harassment, due to aggressive behavior of typical “prey” species rather than using the language of predation.

Line 69 – rather than “understand aspects of biodiversity”, I think these observations “enrich understanding of interactions among the biodiversity” in the Ross Sea region.

Reviewer 2 Report

This ms is reporting that Adelie penguins showed interest in breeding skuas and accidentally broke an egg.

Authors use the word "attacked" in the title but was it really an attack? 

Methods need to be detailed. What was the period of camera monitoring of skua nests? How many nests were monitored and how many times the Adelie penguins were captured? Are there any observation of penguins ignoring skuas?

The interpretation is reasonable but the conclusion (the last sentence) is a bit far fetched. How it could "be helpful to understand aspects of marine biodiversity and ecosystem"? Why "particularly at Ross Sea region"? 

L38, December 19th could be still during the incubation. Do authors have any information about the chronology of the breeding Adelie penguins in this region?

L57, Are they really non-breeders? How often the Adelie penguins were observed? Is it be possible that it was a common route for Adelie penguins passage. Any information about the foraging area of the penguins breeding at Adelie Cove?

L64-65, This sentence is not clear. Adelie penguins' eggs and chicks are the prey of skuas but not adult penguins. 

L67-70, Is it? 
